# A Brief Executive Language Screen for Frontal Aphasia

**DOI:** 10.3390/brainsci11030353

**Published:** 2021-03-10

**Authors:** Gail A. Robinson, Lucy Shi, Zoie Nott, Amelia Ceslis

**Affiliations:** 1Neuropsychology Research Unit, School of Psychology, The University of Queensland, Brisbane, QLD 4072, Australia; lucy.shi@uq.net.au (L.S.); z.nott@uq.edu.au (Z.N.); a.ceslis@uq.edu.au (A.C.); 2Queensland Brain Institute, The University of Queensland, Brisbane, QLD 4072, Australia

**Keywords:** executive functions, aphasia screening, language test, initiation, inhibitory control, selection, propositional language, verbal fluency

## Abstract

Aphasia assessment tools have primarily focused on classical aphasia type and severity, with minimal incorporation of recent findings that suggest a significant role of executive control operations in language generation. Assessment of the interface between language and executive functions is needed to improve detection of spontaneous speech difficulties. In this study we develop a new *Brief Executive Language Screen* (BELS), a brief tool specifically designed to assess core language and executive functions shown to be involved in spontaneous generation of language. Similar to other measures of aphasia, the BELS assesses articulation and core language skills (repetition, naming and comprehension). Unique additions to the BELS include assessments of spontaneous connected speech, word fluency (phonemic/semantic) and sentence completion (verbal initiation, inhibition and selection). One-hundred and eight healthy controls and 136 stroke patients were recruited. Confirmatory factor analysis was used to determine construct validity and logistic regression was used to evaluate the discriminative validity, informing the final version of the BELS. The results showed that the BELS is sensitive for articulation and nominal language deficits, and it measures executive aspects of spontaneous language generation, which is a hallmark of frontal dynamic aphasia. The results have encouraging theoretical and practical implications.

## 1. Introduction

Spoken language is fundamental for the communication of ideas and it is argued to be uniquely human. A number of core language skills including naming, reading, repetition and comprehension comprise nominal language abilities, which map onto traditional aphasia classifications (e.g., Broca’s or Wernicke’s aphasia [1]). By contrast, propositional language is spontaneous or voluntary speech that is novel to any given context and it is used to relay information, convey emotion, direct action and tell stories [2]. It is now widely acknowledged that executive functions and other cognitive processes are involved in generating ideas for propositional language [3,4,5]. However, aphasia assessment tools rarely focus on propositional language and this ‘non-language’ aspect. Here we develop a novel *Brief Executive Language Screen (BELS)* designed to assess both nominal language and executive control processes integral for propositional language.

Theoretical models for producing spoken language generally include three main stages: (1) conceptualization; (2) linguistic formation; and (3) overt articulation (e.g., [6,7,8,9,10,11]). The first *preverbal* conceptualization stage is where messages or ideas are generated, which comprises a communicative intent and conceptual plan [12]. This stage represents the interface between language and cognition, and necessarily involves other cognitive processes such as autobiographical memory, executive control and attentional functions [2]. Notably, a number of models acknowledge that executive functions are involved in transforming an idea into spoken language (e.g., [12,13]). The second stage involves transformation of the preverbal message into language or a linguistic structure. This is achieved by matching concepts to items in the mental lexicon and then devising a ‘phonological score’, which contains grammar and a sentence structure [12]. At the third stage, the phonological score is phonetically encoded for motor articulation, which results in overt speech production [6,7,8,10,11].

### 1.1. Executive Functions and Language Generation

Although hinted in spoken language models, propositional language was discussed as a complex goal-directed behavior by Alexander [14], who explicitly mapped out the role of three specific executive functions in conceptualization or generating ideas; energization, task-setting and monitoring. These key processes enable the speaker to initiate and sustain their attention on the intended focus (energization), decide what ideas are relevant to the focus (task-setting) and check whether the ideas produced are consistent with the focus (monitoring). These executive functions are included in other spoken language models (e.g., [11]), with their role in propositional language reviewed in detail by Barker and colleagues [2].

Directly relevant are investigations of dynamic aphasia, an acquired language disorder that is characterized by profoundly reduced propositional language, despite well-preserved nominal language skills [15]. That is, patients with dynamic aphasia can name, repeat, read and comprehend but they rarely initiate conversation or use language spontaneously to communicate thoughts and ideas (e.g., [16,17,18]). Since Luria and Tsvetkova [19] first provided a theoretical explanation for dynamic aphasia, several accounts for this disproportionately reduced propositional speech have been proposed. Current evidence suggests impairment to one of three mechanisms may underpin the pattern observed in dynamic aphasia; (i) an idea generation deficit in which few conceptual ideas are initiated, (ii) a selection deficit in which there is an inability to select one idea from amongst many competing ideas, and iii. a sequencing deficit where ideas become stuck and repeated [20,21,22,23,24]. The latter sequencing deficit implies poor monitoring and/or inhibitory control to produce inappropriate ideas (e.g., [25,26]). These mechanisms can be conceptualized within the Alexander [14] framework; namely, energization is critical for initiating and sustaining idea generation, task-setting is key for deciding or selecting ideas to focus on and produce, and monitoring is key for sequencing ideas and inhibitory control. Moreover, these executive functions have been associated with the prefrontal cortex; specifically, the superior medial, left lateral and right lateral frontal regions, respectively [14,20,21,22,23,24,27,28]. The BELS incorporates the executive components of initiation, selection and inhibition, in addition to eliciting a sample of spontaneous speech.

### 1.2. Aphasia Assessment Tools

Although assessment of language disorders can be traced back to the 1880′s, standardized measures only became available and used in clinical settings since about 1970 [29]. Examples of widely used tools are the Boston Diagnostic Aphasia Examination (BDAE) [30], the Western Aphasia Battery (WAB) [31] and the Aachen Aphasia Test (AAT) [32]. These *symptom-based* tools assess core language components such as reading, comprehension, articulation, naming and repetition, in different modalities (e.g., spoken and written), typically resulting in a profile that corresponds to classical aphasia syndromes, such as Broca’s aphasia. By contrast, another type of tool assesses *functional communication* in everyday settings (e.g., Functional Communication Profile [33], the Communicative Abilities in Daily Living [34], the Pragmatics Profile of Communication skills in Adults [35]). The main goal of functional communication tools is to ascertain the ability of an individual to convey their main message in everyday life. A third type of approach is *theoretically based* on cognitive neuropsychological models. For example, the Psycholinguistic Assessment of Language Processing in Aphasia (PALPA) [36] assess psycholinguistic properties known to influence processes like single word reading or repetition. Another example is the Comprehensive Aphasia Test (CAT) [29], which assesses language and functional pragmatic components. In summary, current aphasia assessment tools typically assess symptoms, function or both. However, a significant gap is in the assessment of other cognitive processes like executive functions that are crucial for the initial conceptualization stage of propositional language; that is, generating ideas to be communicated.

### 1.3. The Current Study

The aim of the current study was to develop an abbreviated aphasia screening tool to assess the cognitive components of language generation, based on contemporary theoretical models of spoken production that incorporate three broad stages; that is, conceptualization, linguistic formulation and articulation. In addition, the *Brief Executive Language Screen (BELS)* is uniquely designed to identify executive language deficits in neurological disorders including acquired brain injury patients (brain tumor and stroke) and neurodegenerative disorders (e.g., frontotemporal dementia, Alzheimer’s disease, motor neurone disease).

The BELS comprises 11 subscales that assess propositional language (spontaneous speech), nominal language functions (repetition, naming, comprehension, reading), and executive functions that are known to be key for the conceptualization stage of language generation (initiation, selection and inhibition). It also includes a screen of oral apraxia, which assesses motor speech articulation difficulties, and incidental verbal memory. The BELS is a short screening tool (~15–20 min) that can be administered in a hospital setting at bedside.

Specifically, the aims were to:Assess the construct validity and reliability (internal consistency) of the BELS within a three-factor structure model comprising Propositional Language, Nominal Language and Oromotor Function that broadly map onto the Conceptualization, Linguistic Formulation and Articulation stages of spoken language, respectively. It was hypothesized that all items would fit adequately into a three-factor model, demonstrating internally consistent variance within the model.Determine the practical utility of the BELS as an assessment tool for aphasia (discriminant validity). The study aimed to examine the sensitivity of each subtest by distinguishing the level of performance between healthy controls and acute stroke patients. That is, it was hypothesized that lower scores on the BELS would predict stroke status, whereas higher scores (likely close to ceiling) would predict performance from the healthy controls.

## 2. Materials and Methods

### 2.1. Participants

One hundred and eight healthy controls (18–88 years) were recruited from the community, or via The University of Queensland participant scheme. The healthy controls did not have a neurological or psychiatric diagnosis and English was their first language. We recruited 142 stroke patients (18–90 years) from the Princess Alexandra and Royal Brisbane and Women’s Hospitals. Inclusion criterion for the stroke group were (1) first-time stroke diagnosis confirmed by computed tomography (CT) or magnetic resonance imaging (MRI) and (2) within eight weeks of hospital admission due to the stroke. The exclusion criteria were having experienced a previous stroke or transient ischaemic attack (TIA), over the age of 90 years, non-English speaking or the presence of other neurological and/or psychiatric diagnoses. The stroke patients were unselected in that they were not recruited based on the presence of aphasia or lesion site (e.g., left hemisphere). Six stroke patients were excluded due to multiple bilateral lesions (*n* = 3) or technical issues with speech recordings (*n* = 3), resulting in the inclusion of 136 stroke patients (87% cortical: 43% left hemisphere, 44% right hemisphere). Stroke patients were assessed within eight weeks of a stroke at bedside or at home following discharge. As shown in Table 1, the stroke group was significantly older than healthy controls (Mann–Whitney *U* = 8574.50, *p* < 0.05) and had a lower mean number of years of education (*U* = 4448.00, *p* < 001) and level of estimated premorbid intelligence (*U* = 4621.50, *p* < 0.001), as ascertained by the National Adult Reading Test (NART; [37]). In addition, to screen for significant visual impairment that could confound visual-based language tasks, the Incomplete Letters Test of visual perception [38] was given. Although the groups differed (*U* = 3034, *p* < 0.001), no participant performed below the clinical 5th percentile cut-off for impairment. All participants gave informed written consent. Ethical clearance was provided by the University of Queensland (UQ) Human Research Ethics Committee and the Metro South and Metro North Health Human Research Ethics Committees.

### 2.2. Measures

#### Brief Executive Language Screen (BELS)

The BELS evolved from the initial development and discriminant validity stage to the final validity stage. The main change to the final version was to the executive function component of the BELS. Specifically, an inhibition component was added (subtest 9 and 10 below) and the sentence generation subtest was deleted due to redundancy with the sentence completion subtest, which contains all three key executive components (initiation, selection, inhibition). The final BELS comprises 11 subtests (see Appendix A), which were administered in the following order:1.Spontaneous Speech-Participants were given 1 min and asked to “Describe what you see in the picture”. For the initial BELS, a black and white street scene was presented (Figure 1a). The final BELS included a novel black and white Australian Beach Scene (Figure 1b [24]). In a second “Goal” condition, the standard Cookie Theft Scene [30] was presented with the instruction to “talk continuously for one minute about what you see” (for details see [39]). Speech samples were transcribed and scored for quantity (words/minute) and quality (prosody, grammar, errors) [23,24].2.Oral Apraxia—Participants were verbally instructed to perform five actions comprising orofacial movements (coughing, blowing out a match, clucking, whistling, puffing up cheeks). A score of 2 was given for correct execution, 1 for effortful execution or imitation and 0 if unable to complete the action (maximum score of 10). A score below 10 indicates a degree of abnormality in oromotor function.3.Sentence Repetition—Participants were asked to repeat five sentences between three to five words in length. One point was awarded for correct repetition without errors (maximum score of 5).4.Oral Naming—Participants were asked to name 10 line drawings of objects presented on a single A4 sheet of paper (tuning fork, dolphin, harp, saxophone, tiara, koala, wrench, caterpillar, celery, sunflower). Correct responses were given 1 point (maximum score of 10).5.Word Repetition—The 10 Oral Naming items were orally presented and participants were asked to repeat each word. Correct responses were given 1 point (maximum score of 10).6.Comprehension—The A4 stimulus sheet containing the 10 objects used for Oral Naming was presented. The examiner randomly named each item and asked the participant to point to the corresponding picture. Correct responses were given 1 point (maximum score of 10).7.Action Naming—Participants were presented with 5 line drawings depicting actions on an A4 sheet and asked to name the action in the present and past tense (shoot/shot, dig/dug, drink/drank, swim/swam, bite/bit). Correct responses were given 1 point (maximum score of 10).8.Word Fluency—Participants completed a phonemic and semantic word fluency task. They were given a letter (S) or category (animals) cue and asked to generate as many words as possible in 1 min without repeating items and, for the phonemic task, excluding numbers, proper nouns, or the same word with different endings [39]. A second “Goal” condition was included with participants instructed to provide 20% more than what they produced for the standard word fluency tasks (for details see [40]). For the Goal condition the cues differed (letter–B; category–fruit/vegetables).9.Verbal Generation. The initial BELS comprised two tasks to measure verbal initiation and selection; sentence completion and sentence generation, based on previously reported tasks [21,22,39]. The final BELS comprises only the sentence completion task with an additional inhibition condition.
∗*Sentence Completion*-This subtest is comprised of two parts; initiation and inhibition. Participants were orally presented with a sentence frame omitting the final word and asked to produce one word that completes the sentence meaningfully (initiation) or that is unconnected in any way (inhibition) (based on [41]). The stimuli comprise 5 high constraint sentence frames (i.e., a dominant response is available–low selection demands) and 5 low in constraint (i.e., many responses are available–high selection demands) [21,22,42]. Thus, as low constraint items demand greater selection than high constraint items, it is expected that performance will be poorer for the low constraint items.∗*Sentence Generation*–This subtest requires participants to generate a meaningful sentence when orally presented with a single word that is either high in constraint (Proper Nouns) or low in constrain (high frequency words) (for details see [43,44]). As generating a sentence from a high frequency word demands greater selection than from a proper noun, performance is expected to be poorer for high frequency words, which elicit many competing ideas.


For both tasks, response time and number correct were recorded for all conditions. One point was given for each correct response, within the allowed 20 s, with a maximum of 5 per condition and 20 in total.
10.Motor Go-No Go task—Based on Luria’s rhythm tapping task [45], participants are asked to execute a sequence of one or two taps with their hand in response to the examiner’s tapping. In the first trial (Copy), participants tap once in response to one tap from the examiner, and twice in response to two taps from the examiner. In the second trial (Reverse), participants are asked to respond in an opposite manner to the examiner. That is, when the examiner taps once, the participant taps twice, and when the examiner taps twice, the participant taps once. For the first Copy trial, 1point was awarded if able to execute the entire rhythm. For the second Reverse trial, 2 points were awarded if the participant was able to execute the entire rhythm, one point if able to execute part of the rhythm, and no points if they were unable to execute any of the rhythm. The task was discontinued if the participant was unable to complete the practice taps in each trial (maximum score of 3).11.Memory—Participants were asked to recall the 10 items presented in three previous subtests (Oral Naming, Word Repetition, Comprehension). Correct responses were given 1 point (maximum score of 10).

### 2.3. Statistical Analyses

For construct validity (Aim 1), confirmatory factor analysis (CFA) was implemented in R version 3.2.1 [46], package extension *Lavaan* 0.5 to 18 [47]. As most of the BELS subtests were not normally distributed, where relevant, statistical procedures robust to nonparametric data were used (e.g., maximum-likelihood estimation with robust standard errors, Satorra–Bentler-scaled test statistic). For comparison of changes in CFA models, the following robust statistics were evaluated: χ^2^/df ratios [48]), comparative fit indices (CFI; [49]), root-mean-square error of approximation (RMSEA) [50], standardized root-mean-square residual (SRMR) [50], and Akaike information criterion (AIC; [51]). The aim was to assess if, and how well, the variance within the BELS subtests fit together. For this analysis, two models were tested and compared for fit, a unifactor model and a three-factor model of language. The three-factor model comprised the components of Propositional ‘executive’ Language, Nominal Language and Oromotor Function that broadly map onto the generally agreed Conceptual Preparation, Linguistic Formulation and Articulation stages of spoken language, respectively.

To ascertain the internal consistency (reliability), raw scores were first transformed into standardized z-scores. As the scores were recorded in different units of measure (e.g., number correct, words/minute, response time), data transformation allowed for direct and more accurate analyses of the variance amongst these subscales. Cronbach’s Alpha analyses were run for configurations of items that corresponded with the theoretical components. A full scale analysis included all subtests, a nominal language analysis included subtests that assess core language skills (i.e., repetition, object naming, action naming), and the propositional language analysis included subtests that assessed executive language components (i.e., spontaneous speech, word fluency, sentence completion, sentence generation).

To ascertain the discriminative validity of the BELS (Aim 2), logistic regression analyses were conducted using SPSS version 24. The aim was to assess if each subtest can discriminate between the scores of the healthy controls and stroke patients.

## 3. Results

### 3.1. Healthy Control BELS Normative Data

The normative data for the BELS subtest scores for the healthy group is summarized in Table 2. As the distribution of subtest raw scores was not normal and some scores had a limited range, which often occurs in clinical research, measures of central tendency and dispersion were depicted by the median and interquartile ranges (IQR), respectively. In addition, cut-off scores for impairment were set at the 5th percentile.

### 3.2. Stroke Patient Group BELS Scores

A summary of the BELS subtest scores for the stroke group are shown in Table 3. The percentage of patients impaired on each subtest ranged from ~5–50%. For the nominal language subtests, ~25% of stroke patients were impaired for sentence repetition, ~10% for object naming, ~50% for action naming and only ~5% for word repetition and comprehension, the latter reflecting low task difficulty or ceiling effects. As anticipated, the propositional language subtests detected impairment in ~30–50% of stroke patients across all verbal subtests including spontaneous speech, word fluency, initiation, selection (i.e., Initiation LC vs. HC subtest), and verbal inhibition. Although the motor non-verbal inhibition task detected difficulty in a modest number of patients (~13%), the incidental verbal memory subtest identified 41% of stroke patients as impaired and oral apraxia was detected in half of the stroke patients.

### 3.3. Validation of Theoretical Structure (Construct Validity)

To assess the construct validity of the BELS within a three-factor structure model confirmatory factor analysis (CFA) was used. To reduce the effects of non-normal data for the CFA, maximum-likelihood estimation with robust standard errors, and a Satorra–Bentler-scaled test statistic were used. The comprehension subtest was not included due to lack of variability and only sentence repetition, not word repetition was included. Model fit statistics for both unifactor and 3-factor solutions fell within acceptable to good ranges (see Table 4). Guided by Brown [51] and Carmines and McIver [48], CFA models were compared using changes in χ^2^/df ratios (smaller values reflect improved fit), CFI (values > 0.93 indicated good fit), RMSEA and SRMR (values <0.07 indicated good fit) and AIC (smaller values reflected improved fit). The unifactor and 3-factor models showed comparable fits, where the models yielded similar CFI, RMSEA, SRMR, and AIC statistics. These results demonstrate that the subtests did not differ in the way they fell within a global construct of language. The 3-factor model is illustrated below in Figure 2.

### 3.4. Internal Consistency (Reliability)

To analyze statistics for internal consistency, raw scores were first transformed into standardized z-scores. This was due to the vast discrepancy in the magnitude of the scores, as some subtests were measured in words per minute, where most were scored out of 5 or 10. By using z-scores, the Cronbach’s Alpha statistic more accurately represented the pattern of variation in the scores. The Cronbach’s Alpha for all BELS subtests is high overall (α = 0.861), and slightly higher if the comprehension subtest is excluded (α = 0.869). For the Nominal language subtests, Cronbach’s Alpha fell within the moderate range (α = 0.686). For Propositional language subtests, Cronbach’s Alpha fell within the high range (α = 0.815), demonstrating a consistent pattern of variation in these subtests. In summary, the statistics suggested that all subtest scores appeared to vary consistently within the model.

### 3.5. Discriminant Validity

To formally assess the ability of each subtest to discriminate between healthy and stroke participants (Aim 2), logistic regression analyses were conducted. Each BELS subtest was added to the baseline model holding age, education, memory and visual perception constant as covariates. All subtests, except comprehension, semantic fluency, and sentence generation, were significant predictors of whether the participant belonged to the healthy or stroke group, as indicated by significant *p*-values and the increase in both Nagelkerke’s R^2^ and Percentage of Correct Prediction beyond the baseline model (see Table 5). The lack of discriminative ability for the semantic fluency and sentence generation subtests may be explained by entering memory as a covariate because both subtests involve a memory retrieval component. For instance, semantic fluency requires the retrieval of objects from memory and sentence generation requires information regarding the presented word to be retrieved in order to generate a meaningful sentence. With regard to the comprehension subtest, the insufficient variability in scores and ceiling effect likely impacted the ability to discriminate between groups.

## 4. Discussion

This study presents the *Brief Executive Language Screen* (BELS), which was developed to specifically detect the executive function components that are critical for conceptualization when producing propositional language. The BELS targets the three main stages of spoken language production (e.g., [6,7,8,9,10,11,12], including linguistic formulation that broadly reflects core nominal language skills (e.g., naming, comprehension and repetition) and articulation, in addition to the conceptual or propositional language aspects. By examining all three aspects, the BELS is grounded in current theoretical findings [3,4,5] and provides a unique contribution to aphasia assessment tools, which rarely (if at all) explicitly target executive components (e.g., initiation, selection, inhibition).

### 4.1. Theoretical Factor Structure

Construct validity of the BELS was evaluated with confirmatory factor analysis. As predicted, the proposed factor structure demonstrated good model fit, indicating that the oromotor functioning, nominal language and propositional language components comprised of uniquely different constructs within the data collected from stroke patients. This provides evidence that the BELS reflects the three generally agreed upon broad stages of spoken language production proposed in theoretical frameworks (e.g., [6,7,8,9,10,11,12]). It should be noted that the parsimonious unifactorial model also demonstrated adequate fit when compared to the three-factor model. This result alone suggests that the three-factor model has credibility to explain unique variance in the data, because complex models with a higher number of factors are by default statistically penalized in the modelling process.

### 4.2. Sensitivity of the BELS

The BELS demonstrated practical utility as a sensitive measure of aphasia. The BELS was able to discriminate between the performance of the healthy controls and stroke patients. That is, healthy or stroke group membership was predictable based on BELS subtest scores. Individually, the oromotor function subtest and all nominal language subtests (except for comprehension, which will be discussed below) were able to differentially predict healthy or stroke group status. Similarly, the propositional language ‘executive’ subtests, other than semantic fluency and sentence generation, were able to differentiate between the healthy and stroke groups.

With regard to the semantic fluency and sentence generation subtests, it is likely that the baseline model, which held memory constant, precluded additional discriminatory power of these individual subtests as both implicitly involve memory. Specifically, for semantic fluency participants are asked to generate words from a semantic category like animals, which necessitates the retrieval of exemplars from their personal and general knowledge stores. Similarly, when asked to generate a sentence from a single word like ‘Paris’ or ‘table’, one first retrieves knowledge from memory and then constructs a meaningful sentence. The hypothesis that the lack of discriminative power of these two subtests is explained by an artefact of adding memory to the model baseline is supported by the fact that between 24–40% of stroke patients were impaired on these subtests, based on the normative data (i.e., 5th percentile cut-off detailed in Table 2). Thus, 39% of stroke patients were impaired on the semantic fluency subtest. As research has demonstrated that phonemic and semantic fluency measure both overlapping and distinct processes (e.g., [27]), this subtest was retained in the final BELS. In a similar fashion, 24–30% of stroke patients were impaired on the sentence generation subtest. However, the task demands of this subtest overlap with the sentence completion subtest (i.e., verbal initiation and selection). In addition, recent evidence suggests that the sentence completion task elicits a greater selection effect than sentence generation [44]. Thus, including both the sentence completion and sentence generation subtests is redundant. The final version of the BELS only retained the adapted sentence completion subtest, which also keeps the BELS brief. At the same time an inhibition component was added to the sentence completion subtest for the final version of the BELS. Thus, sentence completion subtest includes three executive functions in one task (i.e., verbal initiation, selection and inhibition), which reduces the noise in the data and results by not having separate tasks that assess each executive function separately (as discussed in [41,52]).

Across all BELS subtests, 5–55% of stroke patients were impaired on individual subtests, compared to the normative data (as shown in Table 3). High rates of impairment were detected for oral apraxia and the nominal language subtests of sentence repetition and action naming, with a lower rate for object naming. Only a small percentage of patients were impaired for comprehension, due to ceiling effects (discussed below), and word repetition, which is expected compared to sentence repetition due to greater task difficulty with increasing length (multiple words) and the added component of grammar (syntactic structure). For the propositional language subtests, a substantial percentage (~30–50%) of stroke patients had impairments on verbal subtests including spontaneous speech, word fluency, initiation, selection (i.e., Initiation LC vs. HC subtest), and verbal inhibition, indicating sensitivity of the propositional language subtests. A strength of the BELS is that it contains other subtests including verbal memory, which was also able to detect impairment in 41% of stroke patients. A lower impairment rate of ~13% was detected by the motor non-verbal inhibition task, indicating that severe rather than subtle impairments are detected by this subtest. Overall, these results are consistent with the prevalence rates of oral apraxia, language and executive function deficits reported in acute stroke (e.g., [53,54]). In fact, one seminal study documented a disorder in executive function, abstract reasoning, verbal memory and/or language in ~60–70% of patients in the very early stage post-stroke (~8 days), which is similar to the current stroke sample that is on average 7.3 days post-stroke [52]. The BELS has the advantage of containing multiple components in a single brief screen that can be administered at bedside.

### 4.3. Future Adaptations and Implications

Finally, to consider the comprehension subtest in light of being unable to discriminate between the healthy and stroke groups. To investigate the executive components of conceptualization critical for propositional language, it is necessary to have sufficient core language skills intact; that is, good repetition, naming and comprehension as observed even in the severe form of dynamic aphasia [15,16,17,18,19,20,21,22,23,24]. Otherwise, any propositional language deficit may be underpinned by core nominal difficulties. On the other hand, items need to be sufficiently sensitive to detect any difficulties. The items used for the object naming, word repetition, comprehension (and memory) subtests were not common but they were drawn from a number of animate and inanimate categories (e.g., dolphin, harp). This likely increased their distinctiveness in a spoken word-picture matching task, such as the comprehension subtest. This is reflected in the fact that ~95% of participants performed at ceiling, meaning it is ‘easy’ and lacks discriminative power, which precluded its’ inclusion in most analyses. Nevertheless, this subtest contributed to the three learning trials for the incidental verbal memory test, which is surprisingly one of the most sensitive subtests. To overcome this limitation there are two possibilities: one, to add a number of uncommon or low frequency items from the same categories to decrease the distinctiveness; and two, add a sentence comprehension subtest using the objects to ascertain both semantics and grammar. With respect to the latter, sentences could range from simple (e.g., Point to the harp) to complex (e.g., Point to the sunflower after you point to the dolphin, or Point to the object two spots below, and to the left of the harp). In this way, the subtest would show increased sensitivity to comprehension errors and use the same items, which retains the learning trials for the incidental verbal memory subtest.

The BELS uniquely identifies executive language deficits, which have been referred to as ‘aphasia without aphasia’ or as underpinning social communication difficulties [15], which have been observed in many neurological disorders. For instance, dynamic aphasia that is characterized by severely reduced propositional language in the context of well-preser4ved nominal and orofacial function, has been reported in patients with brain tumors [16,21], traumatic brain injury [19], parkinsonian disorders [20,24], frontotemporal dementia [17,22], as well as stroke [18]. More importantly, propositional language impairments have been reported in patients without significant aphasia (e.g., stroke [26,55], motor neurone disease [56]).

## 5. Conclusions

The current study presents a novel brief language assessment tool that is grounded within our current theoretical understanding of cognitive processes that contribute to spoken language generation beyond a single word. It has become increasingly recognized that propositional language generation lies at the interface between cognition and language, with goal-directed behavior guiding concept formation [14,57]. The *Brief Executive Language Screen* (BELS) targets propositional and nominal language, along with oromotor functioning, with a unique aspect explicitly incorporating the executive components of initiation, selection and inhibition. Within the preliminary analyses, the BELS was found to achieve construct validity and reliability, and subtest-level sensitivity to the presence of impaired performance in an acute stroke population. Overall, the BELS provides a novel, accessible, and sensitive tool to detect executive language impairments. This will improve diagnosis and subsequently targeted treatment of language impairments in a wide range of neurological disorders.

## Figures and Tables

**Figure 1 brainsci-11-00353-f001:**
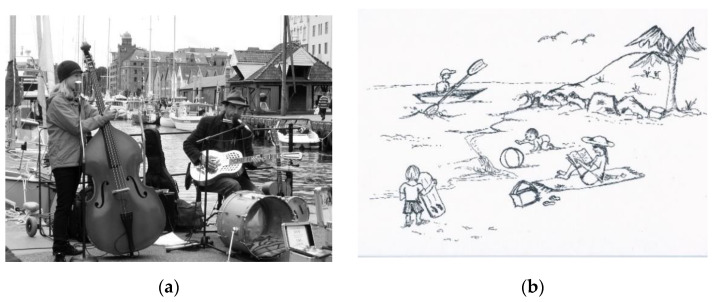
Complex scene description pictures for the BELS spontaneous speech subtest: (**a**) Buskers scene (initial BELS); and (**b**) Australian Beach Scene (final BELS).

**Figure 2 brainsci-11-00353-f002:**
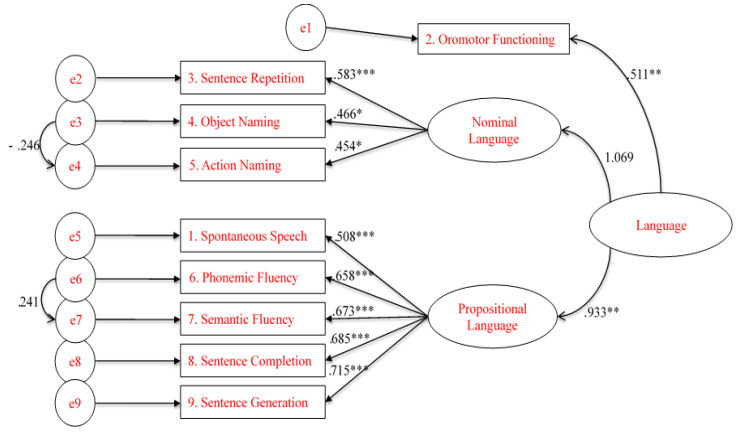
Schematic representation of the 3-factor Confirmatory Factor Analysis model (* *p* < 0.05, ** *p* < 0.01, *** *p* < 0.001).

**Table 1 brainsci-11-00353-t001:** Demographics for Participants (Healthy and Stroke groups).

	Healthy				Stroke			
	*N*	Mean (*SD*)	Median	Range	*N*	Mean (*SD*)	Median	Range
Gender (% Female)	108	54.6%			136	40.4%		
Age (years)	108	53.37 (22.91)	61	18–89	136	61.57 * (16.49)	64	22–91
Education (years)	103	13.50 (2.91)	14	7–20	136	11.51 ** (2.95)	12	7–19
Chronicity (days)	-	-		-	135	7.32 (11.28)	4	1–109
NART-estimated IQ	104	107.26 (8.83)	110	80–127	121	97.55 ** (10.76)	5	75–120
Visuoperception/20	94	19.73 (.64)	20	16–20	111	18.59 * (2.51)	19	0–20

*Note:* NART = National Adult Reading Test. ***
*p* < 05; ** *p* < 001.

**Table 2 brainsci-11-00353-t002:** Normative Data and 5th Percentile Cut Off Scores for Impairment for BELS Subtests.

Subtest (Measure or Maximum Score)	*N*	Mean (SD)	Median	Min	Max	25th %Tile	75th %Tile	5th %Tile
OROMOTOR FUNCTION								
Oral Apraxia/10	106	9.88 (0.43)	10	8	10	10	10	9
NOMINAL LANGUAGE								
Sentence Repetition/5	106	4.97 (0.17)	5	4	5	5	5	5
Word Repetition/10	106	10 (0.00)	10	10	10	10	10	10
Object Naming/10	106	9.64 (0.81)	10	7	10	10	10	8
Action Naming/10	106	9.73 (0.80)	10	6	10	10	10	8
Comprehension/10	106	10 (0.00)	10	10	10	10	10	10
PROPOSITIONAL LANGUAGE/EXECUTIVE FUNCTIONS								
**Spontaneous Speech:** Beach Scene Description (wpm)	42	126.07 (34.65)	120	61	207	103.00	150.00	64
*Buskers Scene Description (wpm)*	*60*	*109.85 (34.63)*	*111*	*34*	*182*	*85.25*	*136*	*50*
**Word Fluency:**								
Phonemic ‘S’ (wpm)	105	15.81 (5.36)	15	4	40	12.50	19.50	9
Semantic Animals (wpm)	42	22.45 (4.98)	21.50	11	32	19	27	15
Goal Phonemic B (wpm)	41	15.46 (5.01)	15	5	26	11.50	18.50	10
*Semantic Fruit/Veg. (wpm)*	*63*	*20.11 (4.85)*	*20*	*6*	*35*	*16*	*24*	*12*
Goal Semantic Fruit/Veg. (wpm)	41	21.02 (4.36)	22	14	34	18	24	14
**Sentence Completion:**								
Total Initiation/10	105	9.59 (0.68)	10	7	10	9	10	8
HC (Low selection)/5	105	4.99 (0.10)	5	4	5	5	5	5
LC (High selection)/5	105	4.60 (0.67)	5	2	5	4	5	3
Inhibition/10	42	7.26 (2.60)	8	1	10	5	10	3
***Sentence Generation:***								
*Total Generation/10*	*62*	*9.61 (0.82)*	*10*	*6*	*10*	*9.75*	*10*	*8*
*Proper Nouns/5*	*62*	*4.74 (0.57)*	*5*	*3*	*5*	*5*	*5*	*3*
*High Frequency Words/5*	*62*	*4.87 (0.38)*	*5*	*3*	*5*	*5*	*5*	*4*
**Nonverbal Inhibition:**								
Motor Go-No Go: Copy/1	41	0.88 (0.33)	1	0	1	1	1	0
Motor Go-No Go: Reverse/2	42	1.81 (0.46)	2	0	2	2	2	1
MEMORY								
Incidental verbal memory/10	102	5.77 (1.88)	6	1	10	4	7	3

Note: WPM = words per minute; Subtests in italics are not included in the final BELS.

**Table 3 brainsci-11-00353-t003:** Descriptive Statistics and Percentage of Stroke Patients Impaired for the BELS subtests.

Subtest (Measure or Maximum Score)	N	Mean (SD)	Median	Min	Max	25th %Tile	75th %Tile	Percentage of Patients ≤5th %Tile
OROMOTOR FUNCTION								
Oral Apraxia/10	136	9.01 (1.33)	9.50	3	10	8	10	50.0%
NOMINAL LANGUAGE								
Sentence Repetition/5	133	4.48 (1.07)	5	0	5	4	5	27.8% ^
Word Repetition/10	135	9.89 (.68)	10	3	10	10	10	4.4% ^
Object Naming/10	135	9.52 (1.20)	10	0	10	9	10	8.9%
Action Naming/10	107	7.79 (2.02)	8	2	10	7	9	55.1%
Comprehension/10	136	9.95 (0.25)	10	8	10	10	10	4.4% ^
PROPOSITIONAL LANGUAGE/EXECUTIVE FUNCTIONS								
**Spontaneous Speech:** *Buskers Scene Description (wpm)*	*136*	*65.78 (39.64)*	*60*	*4*	*199*	*32.25*	*89*	*39.0%*
**Word Fluency:**								
Phonemic ‘S’ (wpm)	134	10.14 (6.54)	9	0	28	5	15	50.7%
Semantic Fruit/Veg. (wpm)	132	14.61 (6.41)	14	0	32	10.25	19.75	38.6%
**Sentence Completion:**								
Initiation/10	125	8.50 (1.68)	9	0	10	8	10	37.6%
HC (Low selection)/5	127	4.87 (.54)	5	0	5	5	5	9.4% ^
LC (High selection)/5	127	3.63 (1.39)	4	0	5	3	5	36.2%
Inhibition/10	58	4.90 (3.35)	5.5	0	10	2	7	37.9%
***Sentence Generation:*** *Generation/10*	*123*	*8.62 (1.95)*	*10*	*2*	*10*	*8*	*10*	*30.9%*
*Proper Nouns/5*	*123*	*4.13 (1.23)*	*5*	*0*	*5*	*4*	*5*	*23.6%*
*High Frequency Words/5*	*124*	*4.49 (0.91)*	*5*	*1*	*5*	*4*	*5*	*29.8%*
**Nonverbal Inhibition:**								
Motor Go-No Go: Copy/1	61	0.98 (0.23)	1	0	1	1	1	3.3%
Motor Go-No Go: Reverse/2	61	1.86 (0.35)	2	1	2	2	2	13.1%
MEMORY								
Incidental verbal memory/10	119	3.56 (2.12)	4	0	8	2	5	41.2%

*Note. ^* Cut-off < 5th%tile as cut-off is full score and designates a pass/fail subtest; *Subtests in italics are not included in the final BELS*.

**Table 4 brainsci-11-00353-t004:** Confirmatory factor analyses for 9-items* BELS scale in unifactor and 3-factor models.

Model	χ^2^ (df)	χ^2^/df	*p*	CFI	RMSEA	SRMR	AIC
Unifactor	28.928 (25)	1.157	0.267	0.979	0.042	0.049	2474.265
Three-Factor	30.004 (23)	1.305	0.149	0.965	0.057	0.048	2477.695

*Note:* Comprehension and Memory subtests not included. CFI = comparative fit indices; RMSEA = root-mean-square error of approximation; SRMR = standardized root-mean-square residual; AIC = Akaike information criterion.

**Table 5 brainsci-11-00353-t005:** Logistic regression for discriminating between the Healthy and Stroke Groups.

							95% CI for Odds Ratios		
	*N*	χ^2^	df	∆*p*	*Nagelkerke R^2^*	Percentage Correct	Odds Ratio ^	Lower	Upper
Baseline Model	156	49.974	4	<0.001	0.375	75.00	-	-	-
Apraxia	156	73.244	5	<0.001	0.513	84.60	0.149	0.046	0.477
Baseline Model	154	48.905	4	<0.001	0.372	76.00	-	-	-
Sentence Repetition	154	56.932	5	0.005	0.422	79.20	0.232	0.055	0.988
Baseline Model	156	49.974	4	<0.001	0.375	75.00	-	-	-
Object Naming	156	57.969	5	0.005	0.425	77.60	0.220	0.053	0.918
Baseline Model	156	49.974	4	<0.001	0.375	75.00	-	-	-
Comprehension	156	NA	NA	NA	NA	NA	NA	NA	NA
Baseline Model	156	49.974	4	<0.001	0.375	75.00	-	-	-
Action Naming	156	58.226	5	0.004	0.426	81.40	0.378	0.176	0.813
Baseline Model	151	45.958	4	<0.001	0.362	78.80	-	-	-
Spontaneous Speech	151	61.676	5	<0.001	0.463	78.80	0.977	0.966	0.989
Baseline Model	142	1.711	4	0.789	0.038	61.00	-	-	-
Phonemic Fluency	142	18.16	5	0.003	0.353	71.20	1.251	1.101	1.422
Baseline Model	152	46.901	4	<0.001	0.366	77.00	-	-	-
Semantic Fluency	152	49.113	5	0.137	0.381	80.90	0.935	0.853	1.023
Baseline Model	152	49.563	4	<0.001	0.38	75.70	-	-	-
Sentence Completion	152	56.807	5	0.007	0.426	78.90	0.465	0.245	0.879
Baseline Model	151	51.241	4	<0.001	0.394	76.80	-	-	-
Sentence Generation	151	53.005	5	0.184	0.405	79.50	0.795	0.557	1.135

^ Odds Ratios represented the probability of past stroke (i.e., coded 0 = Healthy; 1 = Stroke).

## Data Availability

The conditions of our ethics approval do not permit public archiving of the data supporting the conclusions of the study. Readers seeking access to this data should contact the corresponding author (G.A.R.). Access will be granted to named individuals in accordance with ethical procedures governing with reuse of sensitive data. Specifically, requestors must complete a formal data sharing agreement and have local ethics approval to obtain the data.

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
