# Peer review of "A Brief Executive Language Screen for Frontal Aphasia"

_brainsci, 2021, doi:10.3390/brainsci11030353_

Round 1

Reviewer 1 Report

Gail et al. develop a new Brief Executive Language Screen (BELS) in the clinic to evaluate core language and executive functions that are involved in the spontaneous generation of language. The discriminative validity shows the BELS is sensitive for articulation and nominal language deficits in stroke patients and healthy controls. These results provide a quick screening tool in a hospital setting at the bedside.

As the authors mentioned in the future adaptation, The items used for the object naming, word repetition, comprehension (and memory) subtests were drawn from a number of animate and inanimate categories (e.g., dolphin, harp). Around 95% of participants performed at the ceiling which means the screen lacks discriminative power. The authors should include the results after adding a number of uncommon/low-frequency items from the same categories or adding a sentence comprehension subtest using the objects to ascertain both semantics and grammar.

Presentation
Overall the manuscript is well written and easy to follow, it is filled with a good amount of technical detail. .However, there still have several grammar or typo mistakes (such as Line 135: criteria instead of criterion; Line 416: to instead of Tto, etc.)

Author Response

This reviewer agreed with our future adaptation plan; that is, to include additional items to the object naming, word repetition, comprehension (and memory) subtests. It is beyond the scope of the current study to include this future adaptation and it will be included in future manuscripts.

Regarding the “grammar or typo mistakes” highlighted by this reviewer, we retain ‘criteria’ as there is more than 1, rather than changing this to ‘criterion’. We have now corrected ‘Tto’ on Line 416.

Reviewer 2 Report

The authors developed and presented in this paper BELS, an abbreviated aphasia screening tool designed to assess core language skills and executive functions involved in the spontaneous generation of language. The screening tool incorporated the assessment of spontaneous speech, word fluency, and sentence completion (initiation, inhibition, and selection). The authors conducted a confirmatory factor analysis and logistic regression on a sample of 118 healthy controls and 136 stroke patients. Based on the results, they concluded that BELS is a good instrument to assess articulation and nominal language deficits, and it measures executive aspects of spontaneous language generation.

Comments

The number of keywords seems too large. Five to ten keywords would be adequate.

This is an interesting paper presented as a refinement of a screening tool. My main concern is the relatively small number of healthy control participants and their age variability that goes from 18 to 88 years. In order to get normative data, it would be convenient to enlarge the number of participants and to divide the group into different segments (e.g., young adults, middle-aged adults, and older adults). The same may apply to the stroke patient group

The inclusion of the executive functions in BELS is an interesting addition to this screening tool as cognitive control is clearly involved in language. Executive functions change with age. For example, young adults obtain better scores in executive control tasks than older adults. So, it is important to consider the performance differences in terms of the age group.

Author Response

We realise that there are many keywords. However, we retain these as different terms are used in the literature - e.g., propositional language and spontaneous speech are used interchangeably, and at times verbal fluency is also used to refer to the same phenomenon. We are of the view that including the different terms is necessary.

We thank the reviewer for highlighting the what the optimal healthy control group would consist of – i.e., even spread of ages between 18-89. In hindsight, we agree that this would be convenient and desirable; we will do this in our next study. However, for our primary purpose of validating the tool and theoretical foundation itself, our stroke group of 136 patients and healthy group of 118 are sufficient as our results confirm our 3 -structure model and development of a tool based on this.  

Round 2

Reviewer 2 Report

I still think that the number of terms is high but I understand the position of the authors. I would recommend a larger number of participants for future research on the subject.

Author Response

We have reduced the number of keywords by 50%.